# Temporal Difference Models:
## Model-Free Deep RL for Model-Based Control

**Vitchyr Pong**[*]
University of California, Berkeley
vitchyr@berkeley.edu

**Shixiang Gu**[*]
University of Cambridge
Max Planck Institute
Google Brain
sg717@cam.ac.uk

**Murtaza Dalal**
University of California, Berkeley
mdalal@berkeley.edu

**Sergey Levine**
University of California, Berkeley
svlevine@eecs.berkeley.edu

## Abstract

Model-free reinforcement learning (RL) is a powerful, general tool for learning complex behaviors. However, its sample efficiency is often impractically large for solving challenging real-world problems, even with off-policy algorithms such as Q-learning. A limiting factor in classic model-free RL is that the learning signal consists only of scalar rewards, ignoring much of the rich information contained in state transition tuples. Model-based RL uses this information, by training a predictive model, but often does not achieve the same asymptotic performance as model-free RL due to model bias. We introduce temporal difference models (TDMs), a family of goal-conditioned value functions that can be trained with model-free learning and used for model-based control. TDMs combine the benefits of model-free and model-based RL: they leverage the rich information in state transitions to learn very efficiently, while still attaining asymptotic performance that exceeds that of direct model-based RL methods. Our experimental results show that, on a range of continuous control tasks, TDMs provide a substantial improvement in efficiency compared to state-of-the-art model-based and model-free methods.

## 1 Introduction

Reinforcement learning (RL) algorithms provide a formalism for autonomous learning of complex behaviors. When combined with rich function approximators such as deep neural networks, RL can provide impressive results on tasks ranging from playing games (Mnih et al., 2015; Silver et al., 2016), to flying and driving (Lillicrap et al., 2015; Zhang et al., 2016), to controlling robotic arms (Levine et al., 2016; Gu et al., 2017). However, these deep RL algorithms often require a large amount of experience to arrive at an effective solution, which can severely limit their application to real-world problems where this experience might need to be gathered directly on a real physical system. Part of the reason for this is that direct, model-free RL learns only from the reward: experience that receives no reward provides minimal supervision to the learner.

In contrast, model-based RL algorithms obtain a large amount of supervision from every sample, since they can use each sample to better learn how to predict the system dynamics – that is, to learn the "physics" of the problem. Once the dynamics are learned, near-optimal behavior can in principle be obtained by planning through these dynamics. Model-based algorithms tend to be substantially more efficient (Deisenroth et al., 2013; Nagabandi et al., 2017), but often at the cost of larger asymptotic bias: when the dynamics cannot be learned perfectly, as is the case for most complex problems, the final policy can be highly suboptimal. Therefore, conventional wisdom holds that model-free methods are less efficient but achieve the best asymptotic performance, while model-based methods are more efficient but do not produce policies that are as optimal.

---

[*]denotes equal contribution

Can we devise methods that retain the efficiency of model-based learning while still achieving the asymptotic performance of model-free learning? This is the question that we study in this paper. The search for methods that combine the best of model-based and model-free learning has been ongoing for decades, with techniques such as synthetic experience generation (Sutton, 1990), partial model-based backpropagation (Nguyen & Widrow, 1990; Heess et al., 2015), and layering model-free learning on the residuals of model-based estimation (Chebotar et al., 2017) being a few examples. However, a direct connection between model-free and model-based RL has remained elusive. By effectively bridging the gap between model-free and model-based RL, we should be able to smoothly transition from learning models to learning policies, obtaining rich supervision from every sample to quickly gain a moderate level of proficiency, while still converging to an unbiased solution.

To arrive at a method that combines the strengths of model-free and model-based RL, we study a variant of goal-conditioned value functions (Sutton et al., 2011; Schaul et al., 2015; Andrychowicz et al., 2017). Goal-conditioned value functions learn to predict the value function for every possible goal state. That is, they answer the following question: what is the expected reward for reaching a particular state, given that the agent is attempting (as optimally as possible) to reach it? The particular choice of reward function determines what such a method actually does, but rewards based on distances to a goal hint at a connection to model-based learning: if we can predict how easy it is to reach any state from any current state, we must have some kind of understanding of the underlying "physics." In this work, we show that we can develop a method for learning variable-horizon goal-conditioned value functions where, for a specific choice of reward and horizon, the value function corresponds directly to a model, while for larger horizons, it more closely resembles model-free approaches. Extension toward more model-free learning is thus achieved by acquiring "multi-step models" that can be used to plan over progressively coarser temporal resolutions, eventually arriving at a fully model-free formulation.

The principle contribution of our work is a new RL algorithm that makes use of this connection between model-based and model-free learning to learn a specific type of goal-conditioned value function, which we call a temporal difference model (TDM). This value function can be learned very efficiently, with sample complexities that are competitive with model-based RL, and can then be used with an MPC-like method to accomplish desired tasks. Our empirical experiments demonstrate that this method achieves substantially better sample complexity than fully model-free learning on a range of challenging continuous control tasks, while outperforming purely model-based methods in terms of final performance. Furthermore, the connection that our method elucidates between model-based and model-free learning may lead to a range of interesting future methods.

## 2 PRELIMINARIES

In this section, we introduce the reinforcement learning (RL) formalism, temporal difference Q-learning methods, model-based RL methods, and goal-conditioned value functions. We will build on these components to develop temporal difference models (TDMs) in the next section. RL deals with decision making problems that consist of a state space $\mathcal{S}$, action space $\mathcal{A}$, transition dynamics $P(s' \mid s, a)$, and an initial state distribution $p_0$. The goal of the learner is encapsulated by a reward function $r(s, a, s')$. Typically, long or infinite horizon tasks also employ a discount factor $\gamma$, and the standard objective is to find a policy $\pi(a \mid s)$ that maximizes the expected discounted sum of rewards, $\mathbb{E}_\pi[\sum_t \gamma^t r(s_t, a_t, s_{t+1})]$, where $s_0 \sim p_0$, $a_t \sim \pi(a_t|s_t)$, and $s_{t+1} \sim P(s' \mid s, a)$.

**Q-functions.** We will focus on RL algorithms that learn a Q-function. The Q-function represents the expected total (discounted) reward that can be obtained by the optimal policy after taking action $a_t$ in state $s_t$, and can be defined recursively as following:

$$Q(s_t, a_t) = \mathbb{E}_{p(s_{t+1}|s_t, a_t)}[r(s_t, a_t, s_{t+1}) + \gamma \max_a Q(s_{t+1}, a)]. \tag{1}$$

The optimal policy can then recovered according to $\pi(a_t|s_t) = \delta(a_t = \arg\max_a Q(s_t, a))$. Q-learning algorithms (Watkins & Dayan, 1992; Riedmiller, 2005) learn the Q-function via an off-policy stochastic gradient descent algorithm, estimating the expectation in the above equation with samples collected from the environment and computing its gradient. Q-learning methods can use transition tuples $(s_t, a_t, s_{t+1}, r_t)$ collected from any exploration policy, which generally makes them more efficient than direct policy search, though still less efficient than purely model-based methods.

**Model-based RL and optimal control.** Model-based RL takes a different approach to maximize the expected reward. In model-based RL, the aim is to train a model of the form $f(s_t, a_t)$ to predict the next state $s_{t+1}$. Once trained, this model can be used to choose actions, either by backpropagating reward gradients into a policy, or planning directly through the model. In the latter case, a particularly effective method for employing a learned model is model-predictive control (MPC), where a new action plan is generated at each time step, and the first action of that plan is executed, before replanning begins from scratch. MPC can be formalized as the following optimization problem:

$$a_t = \operatorname*{argmax}_{a_{t:t+T}} \sum_{i=t}^{t+T} r(s_i, a_i) \text{ where } s_{i+1} = f(s_i, a_i) \; \forall \; i \in \{t, ..., t+T-1\}. \qquad (2)$$

We can also write the dynamics constraint in the above equation in terms of an implicit dynamics, according to

$$a_t = \operatorname*{argmax}_{a_{t:t+T}, s_{t+1:t+T}} \sum_{i=t}^{t+T} r(s_i, a_i) \text{ such that } C(s_i, a_i, s_{i+1}) = 0 \; \forall \; i \in \{t, ..., t+T-1\}, \qquad (3)$$

where $C(s_i, a_i, s_{i+1}) = 0$ if and only if $s_{i+1} = f(s_i, a_i)$. This implicit version will be important in understanding the connection between model-based and model-free RL.

**Goal-conditioned value functions.** Q-functions trained for a specific reward are specific to the corresponding task, and learning a new task requires optimizing an entirely new Q-function. Goal-conditioned value functions address this limitation by conditioning the Q-value on some task description vector $s_g \in \mathcal{G}$ in a goal space $\mathcal{G}$. This goal vector induces a parameterized reward $r(s_t, a_t, s_{t+1}, s_g)$, which in turn gives rise to parameterized Q-functions of the form $Q(s, a, s_g)$. A number of goal-conditioned value function methods have been proposed in the literature, such as universal value functions (Schaul et al., 2015) and Horde (Sutton et al., 2011). When the goal corresponds to an entire state, such goal-conditioned value functions usually predict how well an agent can reach a particular state, *when it is trying to reach it*. The knowledge contained in such a value function is intriguingly close to a model: knowing how well you can reach any state is closely related to understanding the physics of the environment. With Q-learning, these value functions can be learned for any goal $s_g$ using the same off-policy $(s_t, a_t, s_{t+1})$ tuples. Relabeling previously visited states with the reward for any goal leads to a natural data augmentation strategy, since each tuple can be replicated many times for many different goals without additional data collection. Andrychowicz et al. (2017) used this property to produce an effective curriculum for solving multi-goal task with delayed rewards. As we discuss below, relabeling past experience with different goals enables goal-conditioned value functions to learn much more quickly from the same amount of data.

## 3 Temporal Difference Model Learning

In this section, we introduce a type of goal-conditioned value functions called temporal difference models (TDMs) that provide a direct connection to model-based RL. We will first motivate this connection by relating the model-based MPC optimizations in Equations (2) and (3) to goal-conditioned value functions, and then present our temporal difference model derivation, which extends this connection from a purely model-based setting into one that becomes increasingly model-free.

### 3.1 From Goal-Conditioned Value Functions to Models

Let us consider the choice of reward function for the goal conditioned value function. Although a variety of options have been explored in the literature (Sutton et al., 2011; Schaul et al., 2015; Andrychowicz et al., 2017), a particularly intriguing connection to model-based RL emerges if we set $\mathcal{G} = \mathcal{S}$, such that $g \in \mathcal{G}$ corresponds to a *goal state* $s_g \in \mathcal{S}$, and we consider distance-based reward functions $r_d$ of the following form:

$$r_d(s_t, a_t, s_{t+1}, s_g) = -D(s_{t+1}, s_g),$$

where $D(s_{t+1}, s_g)$ is a distance, such as the Euclidean distance $D(s_{t+1}, s_g) = \|s_{t+1} - s_g\|_2$. If $\gamma = 0$, we have $Q(s_t, a_t, s_g) = -D(s_{t+1}, s_g)$ at convergence of Q-learning, which means that

$Q(s_t, a_t, s_g) = 0$ implies that $s_{t+1} = s_g$. Plug this Q-function into the model-based planning optimization in Equation (3), denoting the task control reward as $r_c$, such that the solution to

$$a_t = \underset{a_{t:t+T}, s_{t+1:t+T}}{\mathrm{argmax}} \sum_{i=t}^{t+T} r_c(s_i, a_i) \text{ such that } Q(s_i, a_i, s_{i+1}) = 0 \ \forall \ i \in \{t, ..., t+T-1\} \quad (4)$$

yields a model-based plan. We have now derived a precise connection between model-free and model-based RL, in that model-free learning of goal-conditioned value functions can be used to directly produce an implicit model that can be used with MPC-based planning. However, this connection by itself is not very useful: the resulting implicit model is fully model-based, and does not provide any kind of long-horizon capability. In the next section, we show how to extend this connection into the long-horizon setting by introducing the temporal difference model (TDM).

### 3.2 Long-Horizon Learning with Temporal Difference Models

If we consider the case where $\gamma > 0$, the optimization in Equation (4) no longer corresponds to any optimal control method. In fact, when $\gamma = 0$, Q-values have well-defined units: units of distance between states. For $\gamma > 0$, no such interpretation is possible. The key insight in temporal difference models is to introduce a different mechanism for aggregating long-horizon rewards. Instead of evaluating Q-values as discounted sums of rewards, we introduce an additional input $\tau$, which represents the planning horizon, and define the Q-learning recursion as

$$Q(s_t, a_t, s_g, \tau) = \mathbb{E}_{p(s_{t+1}|s_t, a_t)}[-D(s_{t+1}, s_g)\mathbb{1}[\tau = 0] + \max_a Q(s_{t+1}, a, s_g, \tau - 1)\mathbb{1}[\tau \neq 0]]. \quad (5)$$

The Q-function uses a reward of $-D(s_{t+1}, s_g)$ when $\tau = 0$ (at which point the episode terminates), and decrements $\tau$ by one at every other step. Since this is still a well-defined Q-learning recursion, it can be optimized with off-policy data and, just as with goal-conditioned value functions, we can resample new goals $s_g$ and new horizons $\tau$ for each tuple $(s_t, a_t, s_{t+1})$, even ones that were not actually used when the data was collected. In this way, the TDM can be trained very efficiently, since every tuple provides supervision for every possible goal and every possible horizon.

The intuitive interpretation of the TDM is that it tells us how close the agent will get to a given goal state $s_g$ after $\tau$ time steps, *when it is attempting to reach that state in $\tau$ steps*. Alternatively, TDMs can be interpreted as Q-values in a finite-horizon MDP, where the horizon is determined by $\tau$. For the case where $\tau = 0$, TDMs effectively learn a model, allowing TDMs to be incorporated into a variety of planning and optimal control schemes at test time as in Equation (4). Thus, we can view TDM learning as an interpolation between model-based and model-free learning, where $\tau = 0$ corresponds to the single-step prediction made in model-based learning and $\tau > 0$ corresponds to the long-term prediction made by typical Q-functions. While the correspondence to models is not the same for $\tau > 0$, if we only care about the reward at every $K$ step, then we can recover a correspondence by replace Equation (4) with

$$a_t = \underset{a_{t:K:t+T}, s_{t+K:K:t+T}}{\mathrm{argmax}} \sum_{i=t, t+K, ..., t+T} r_c(s_i, a_i)$$
$$\text{such that } Q(s_i, a_i, s_{i+K}, K-1) = 0 \ \forall \ i \in \{t, t+K, ..., t+T-K\}, \quad (6)$$

where we only optimize over every $K^{\text{th}}$ state and action. As the TDM becomes effective for longer horizons, we can increase $K$ until $K = T$, and plan over only a single effective time step:

$$a_t = \underset{a_t, a_{t+T}, s_{t+T}}{\mathrm{argmax}} r_c(s_{t+T}, a_{t+T}) \text{ such that } Q(s_t, a_t, s_{t+T}, T-1) = 0. \quad (7)$$

This formulation does result in some loss of generality, since we no longer optimize the reward at the intermediate steps. This limits the multi-step formulation to terminal reward problems, but does allow us to accommodate arbitrary reward functions on the terminal state $s_{t+T}$, which still describes a broad range of practically relevant tasks. In the next section, we describe how TDMs can be implemented and used in practice for continuous state and action spaces.

## 4 Training and Using Temporal Difference Models

The TDM can be trained with any off-policy Q-learning algorithm, such as DQN (Mnih et al., 2015), DDPG (Lillicrap et al., 2015), NAF (Gu et al., 2016), and SDQN (Metz et al., 2017). During off-policy Q-learning, TDMs can benefit from arbitrary relabeling of the goal states $g$ and the

horizon $\tau$, given the same $(s_t, a_t, s_{t+1})$ tuples from the behavioral policy as done in (Andrychowicz et al., 2017). This relabeling enables simultaneous, data-efficient learning of short-horizon and long-horizon behaviors for arbitrary goal states, unlike previously proposed goal-conditioned value functions that only learn for a single time scale, typically determined by a discount factor (Schaul et al., 2015; Andrychowicz et al., 2017). In this section, we describe the design decisions needed to make practical a TDM algorithm.

## 4.1 Reward Function Specification

Q-learning typically optimizes scalar rewards, but TDMs enable us to increase the amount of supervision available to the Q-function by using a vector-valued reward. Specifically, if the distance $D(s, s_g)$ factors additively over the dimensions, we can train a vector-valued Q-function that predicts per-dimension distance, with the reward function for dimension $j$ given by $-D_j(s_j, s_{g,j})$. We use the $\ell_1$ norm in our implementation, which corresponds to absolute value reward $-|s_j - s_{g,j}|$. The resulting vector-valued Q-function can learn distances along each dimension separately, providing it with more supervision from each training point. Empirically, we found that this modifications provides a substantial boost in sample efficiency.

We can optionally make an improvement to TDMs if we know that the task reward $r_c$ depends only on some subset of the state or, more generally, state features. In that case, we can train the TDM to predict distances along only those dimensions or features that are used by $r_c$, which in practice can substantially simplify the corresponding prediction problem. In our experiments, we illustrate this property by training TDMs for pushing tasks that predict distances from an end-effector and pushed object, without accounting for internal joints of the arm, and similarly for various locomotion tasks.

## 4.2 Policy Extraction with TDMs

While the TDM optimal control formulation Equation (7) drastically reduces the number of states and actions to be optimized for long-term planning, it requires solving a constrained optimization problem, which is more computationally expensive than unconstrained problems. We can remove the need for a constrained optimization through a specific architectural decision in the design of the function approximator for $Q(s, a, s_g, \tau)$. We define the Q-function as $Q(s, a, s_g, \tau) = -\|f(s, a, s_g, \tau) - s_g\|$, where $f(s, a, s_g, \tau)$ outputs a state vector. By training the TDM with a standard Q-learning method, $f(s, a, s_g, \tau)$ is trained to explicitly predict the state that will be reached by a policy attempting to reach $s_g$ in $\tau$ steps. This model can then be used to choose the action with fully explicit MPC as below, which also allows straightforward derivation of a multi-step version as in Equation (6).

$$a_t = \operatorname*{argmax}_{a_t, a_{t+T}, s_{t+T}} r_c(f(s_t, a_t, s_{t+T}, T-1), a_{t+T}) \tag{8}$$

In the case where the task is to reach a goal state $s_g$, a simpler approach to extract a policy is to use the TDM directly:

$$a_t = \operatorname*{argmax}_a Q(s_t, a, s_g, T) \tag{9}$$

In our experiments, we use Equations (8) and (9) to extract a policy.

## 4.3 Algorithm Summary

The algorithm is summarized as Algorithm 1. A crucial difference from prior goal-conditioned value function methods (Schaul et al., 2015; Andrychowicz et al., 2017) is that our algorithm can be used to act according to an arbitrary terminal reward function $r_c$, both during exploration and at test time. Like other off-policy algorithms (Mnih et al., 2015; Lillicrap et al., 2015), it consists of exploration and Q-function fitting. Noise is injected for exploration, and Q-function fitting uses standard Q-learning techniques, with target networks $Q'$ and experience replay (Mnih et al., 2015; Lillicrap et al., 2015). If we view the Q-function fitting as model fitting, the algorithm also resembles iterative model-based RL, which alternates between collecting data using the learned dynamics model for planning (Deisenroth & Rasmussen, 2011) and fitting the model. Since we focus on continuous tasks, we use DDPG (Lillicrap et al., 2015), though any Q-learning method could be used.

The computation cost of the algorithm is mostly determined by the number of updates to fit the Q-function per transition, $I$. In general, TDMs can benefit from substantially larger $I$ than classic

---

**Algorithm 1** Temporal Difference Model Learning

---

**Require:** Task reward function $r_c(s, a)$, parameterized TDM $Q_w(s, a, s_g, \tau)$, replay buffer $\mathcal{B}$

1: **for** $n = 0, ..., N - 1$ episodes **do**
2:     $s_0 \sim p(s_0)$
3:     **for** $t = 0, ..., T - 1$ time steps **do**
4:         $a_t^* = \text{MPC}(r_c, s_t, Q_w, T - t)$         // Eq. 6, Eq. 7, Eq. 8, or Eq. 9
5:         $a_t = \text{AddNoise}(a_t^*)$         // Noisy exploration
6:         $s_{t+1} \sim p(s_t, a_t)$, and store $\{s_t, a_t, s_{t+1}\}$ in the replay buffer $\mathcal{B}$     // Step environment
7:         **for** $i = 0, I - 1$ iterations **do**
8:           Sample $M$ transitions $\{s_m, a_m, s_m'\}$ from the replay $\mathcal{B}$.
9:           Relabel time horizons and goal states $\tau_m, s_{g,m}$         // Section A.1
10:          $y_m = -\|s_m' - s_{g,m}\| \mathbb{1}[\tau_m = 0] + \max_a Q'(s_m', a, s_{g,m}, \tau_m - 1) \mathbb{1}[\tau_m \neq 0]$
11:          $L(w) = \sum_m (Q_w(s_m, a_m, s_{g,m}, \tau_m) - y_m)^2 / M$     // Compute the loss
12:          Minimize($w, L(w)$)         // Optimize
13:         **end for**
14:     **end for**
15: **end for**

---

model-free methods such as DDPG due to relabeling increasing the amount of supervision signals. In real-world applications such as robotics where we care most of the sample efficiency (Gu et al., 2017), the learning is often bottlenecked by the data collection rather than the computation, and therefore large $I$ values are usually not a significant problem and can continuously benefit from the acceleration in computation.

## 5 RELATED WORK

Combining model-based and model-free reinforcement learning techniques is a well-studied problem, though no single solution has demonstrated all of the benefits of model-based and model-free learning. Some methods first learn a model and use this model to simulate experience (Sutton, 1990; Gu et al., 2016) or compute better gradients for model-free updates (Heess et al., 2015; Nguyen & Widrow, 1990). Other methods use model-free algorithms to correct for the local errors made by the model (Chebotar et al., 2017; Bansal et al., 2017). While these prior methods focused on combining different model-based and model-free RL techniques, our method proposes an equivalence between these two branches of RL through a specific generalization of goal-conditioned value function. As a result, our approach achieves much better sample efficiency in practice on a variety of challenging reinforcement learning tasks than model-free alternatives, while exceeding the performance of purely model-based approaches.

We are not the first to study the connection between model-free and model-based methods, with Boyan (1999) and Parr et al. (2008) being two notable examples. Boyan (1999) shows that one can extract a model from a value function when using a tabular representation of the transition function. Parr et al. (2008) shows that, for linear function approximators, the model-free and model-based RL approaches produce the same value function at convergence. Our contribution differs substantially from these: we are not aiming to show that model-free RL performs similarly to model-based RL at convergence, but rather how we can achieve sample complexity comparable to model-based RL while retaining the favorable asymptotic performance of model-free RL in complex tasks with non-linear function approximation.

A central component of our method is to train goal-conditioned value functions. Many variants of goal-conditioned value functions have been proposed in the literature Foster & Dayan (2002); Sutton et al. (2011); Schaul et al. (2015); Dosovitskiy & Koltun (2016). Critically, unlike the works on contextual policies (Caruana, 1998; Da Silva et al., 2012; Kober et al., 2012) which require on-policy trajectories with each new goal, the value function approaches such as Horde (Sutton et al., 2011) and UVF (Schaul et al., 2015) can reuse off-policy data to learn rich contextual value functions using the same data.

TDMs condition on a policy trying to reach a goal and must predict $\tau$ steps into the future. This type of prediction is similar to the prediction made by prior work on multi-step models (Mishra et al., 2017; Venkatraman et al., 2016): predict the state after $\tau$ actions. An important difference is that

multi-step models do not condition on a policy reaching a goal, and so they require optimizing over a sequence of actions, making the input space grow linearly with the planning horizon.

A particularly related UVF extension is hindsight experience replay (HER) Andrychowicz et al. (2017). Both HER and our method retroactively relabel past experience with goal states that are different from the goal aimed for during data collection. However, unlike our method, the standard UVF in HER uses a single temporal scale when learning, and does not explicitly provide for a connection between model-based and model-free learning. The practical result of these differences is that our approach empirically achieves substantially better sample complexity than HER on a wide range of complex continuous control tasks, while the theoretical connection between model-based and model-free learning suggests a much more flexible use of the learned Q-function inside a planning or optimal control framework.

Lastly, our motivation is shared by other lines of work besides goal-conditioned value functions that aim to enhance supervision signals for model-free RL (Silver et al., 2017; Jaderberg et al., 2017; Bellemare et al., 2017). Predictions (Silver et al., 2017) augment classic RL with multi-step reward predictions, while UNREAL (Jaderberg et al., 2017) also augments it with pixel control as a secondary reward objective. These are substantially different methods from our work, but share the motivation to achieve efficient RL by increasing the amount of learning signals from finite data.

## 6 EXPERIMENTS

Our experiments examine how the sample efficiency and performance of TDMs compare to both model-based and model-free RL algorithms. We expect to have the efficiency of model-based RL but with less model bias. We also aim to study the importance of several key design decisions in TDMs, and evaluate the algorithm on a real-world robotic platform. For the model-free comparison, we compare to DDPG (Lillicrap et al., 2015), which typically achieves the best sample efficiency on benchmark tasks (Duan et al., 2016); HER, which uses goal-conditioned value functions (Andrychowicz et al., 2017); and DDPG with the same sparse rewards of HER. For the model-based comparison, we compare to the model-based component in Nagabandi et al. (2017), a recent work that reports highly efficient learning with neural network dynamics models. Details of the baseline implementations are in the Appendix. We perform the comparison on five simulated tasks: (1) a 7 DoF arm reaching various random end-effector targets, (2) an arm pushing a puck to a target location, (3) a planar cheetah attempting to reach a goal velocity (either forward or backward), (4) a quadrupedal ant attempting to reach a goal position, and (5) an ant attempting to reach a goal position and velocity. The tasks are shown in Figure 1 and terminate when either the goal is reached or the time horizon is reached. The pushing task requires long-horizon reasoning to reach and push the puck. The cheetah and ant tasks require handling many contact discontinuities which is challenging for model-based methods, with the ant environment having particularly difficult dynamics given the larger state and action space. The ant position and velocity task presents a scenario where reward shaping as in traditional RL methods may not lead to optimal behavior, since one cannot maintain both a desired position and velocity. However, such a task can be very valuable in realistic settings. For example, if we want the ant to jump, we might instruct it to achieve a particular velocity at a particular location. We also tested TDMs on a real-world robot arm reaching end-effector positions, to study its applicability to real-world tasks.

For the simulated and real-world 7-DoF arm, our TDM is trained on all state components. For the pushing task, our TDM is trained on the hand and puck XY-position. For the half cheetah task, our TDM is trained on the velocity of the cheetah. For the ant tasks, our TDM is trained on either the position or the position and velocity for the respective task. Full details are in the Appendix.

### 6.1 TDMs vs Model-Free, Mode-Based, and Direct Goal-Conditioned RL

The results are shown in Figure 2. When compared to the model-free baselines, the pure model-based method learns learns much faster on all the tasks. However, on the harder cheetah and ant tasks, its final performance is worse due to model bias. TDMs learn as quickly or faster than the model-based method, but also always learn policies that are as good as if not better than the model-free policies. Furthermore, TDMs requires fewer samples than the model-free baselines on ant tasks and drastically fewer samples on the other tasks. We also see that using HER does not lead to

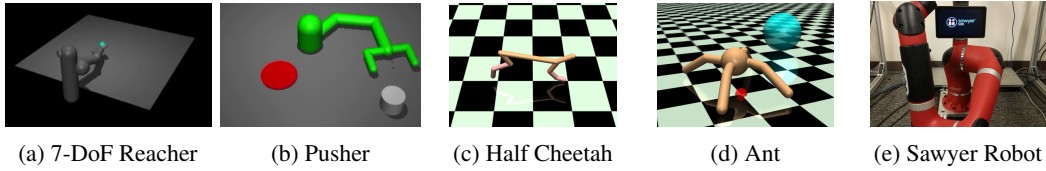

(a) 7-DoF Reacher     (b) Pusher     (c) Half Cheetah     (d) Ant     (e) Sawyer Robot

Figure 1: The tasks in our experiments: (a) reaching target locations, (b) pushing a puck to a random target, (c) training the cheetah to run at target velocities, (d) training an ant to run to a target position or a target position and velocity, and (e) reaching target locations (real-world Sawyer robot).

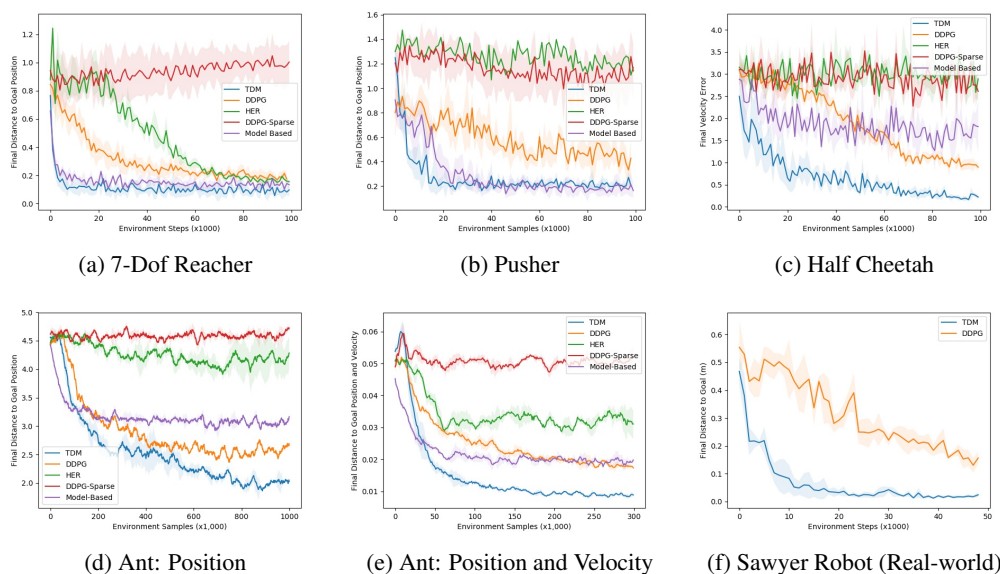

(a) 7-Dof Reacher     (b) Pusher     (c) Half Cheetah

(d) Ant: Position     (e) Ant: Position and Velocity     (f) Sawyer Robot (Real-world)

Figure 2: The comparison of TDM with the baseline methods in model-free (DDPG), model-based, and goal-conditioned value functions (HER - Dense) on various tasks. All plots show the final distance to the goal versus 1000 environment steps (not rollouts). The bold line shows the mean across 3 random seeds, and the shaded region show one standard deviation. Our method, which uses model-free learning, is generally more sample-efficient than model-free alternatives including DDPG and HER and improves upon the best model-based performance.

an improvement over DDPG. While we were initially surprised, we realized that a selling point of HER is that it can solve sparse tasks that would otherwise be unsolvable. In this paper, we were interested in improving the sample efficiency and not the feasibility of model-free reinforcement learning algorithms, and so we focused on tasks that DDPG could already solve. In these sorts of tasks, the advantage of HER over DDPG with a dense reward is not expected. To evaluate HER as a method to solve sparse tasks, we included the DDPG-Sparse baseline and we see that HER significantly outperforms it as expected. In summary, TDMs converge as fast or faster than model-based learning (which learns faster than the model-free baselines), while achieving final performance that is as good or better that the model-free methods on all tasks.

Lastly, we ran the algorithm on a 7-DoF Sawyer robotic arm to learn a real-world analogue of the reaching task. Figure 2f shows that the algorithm outperforms and learns with fewer samples than DDPG, our model-free baseline. These results show that TDMs can scale to real-world tasks.

## 6.2 ABLATION STUDIES

In this section, we discuss two key design choices for TDMs that provide substantially improved performance. First, Figure 3a examines the tradeoffs between the vectorized and scalar rewards. The results show that the vectorized formulation learns substantially faster than the naïve scalar variant. Second, Figure 3b compares the learning speed for different horizon values $\tau_{max}$. Performance degrades when the horizon is too low, and learning becomes slower when the horizon is too high.

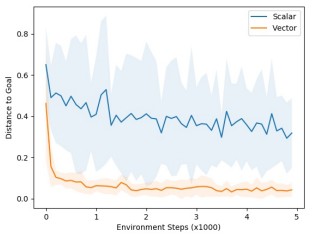 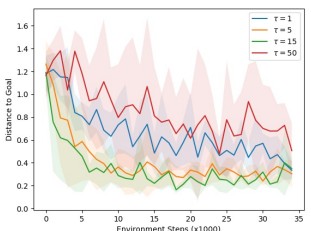

(a) Scalar vs Vectorized TDMs   (b) TDMs with different $\tau_{max}$

Figure 3: Ablation experiments for (a) scalar vs. vectorized TDMs on 7-DoF simulated reacher task and (b) different $\tau_{max}$ on pusher task. The vectorized variant performs substantially better, while the horizon effectively interpolates between model-based and model-free learning.

## 7 CONCLUSION

In this paper, we derive a connection between model-based and model-free reinforcement learning, and present a novel RL algorithm that exploits this connection to greatly improve on the sample efficiency of state-of-the-art model-free deep RL algorithms. Our temporal difference models can be viewed both as goal-conditioned value functions and implicit dynamics models, which enables them to be trained efficiently on off-policy data while still minimizing the effects of model bias. As a result, they achieve asymptotic performance that compares favorably with model-free algorithms, but with a sample complexity that is comparable to purely model-based methods.

While the experiments focus primarily on the new RL algorithm, the relationship between model-based and model-free RL explored in this paper provides a number of avenues for future work. We demonstrated the use of TDMs with a very basic planning approach, but further exploring how TDMs can be incorporated into powerful constrained optimization methods for model-predictive control or trajectory optimization is an exciting avenue for future work. Another direction for future is to further explore how TDMs can be applied to complex state representations, such as images, where simple distance metrics may no longer be effective. Although direct application of TDMs to these domains is not straightforward, a number of works have studied how to construct metric embeddings of images that could in principle provide viable distance functions. We also note that while the presentation of TDMs have been in the context of deterministic environments, the extension to stochastic environments is straightforward: TDMs would learn to predict the *expected* distance between the future state and a goal state. Finally, the promise of enabling sample-efficient learning with the performance of model-free RL and the efficiency of model-based RL is to enable widespread RL application on real-world systems. Many applications in robotics, autonomous driving and flight, and other control domains could be explored in future work.

## 8 ACKNOWLEDGMENT

This research was supported by the Office of Naval Research and the National Science Foundation through IIS-1614653 and IIS-1651843.

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

## A   EXPERIMENT DETAILS

In this section, we detail the experimental setups used in our results.

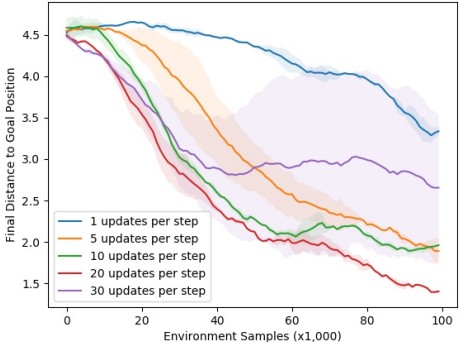

Figure 4: TDMs with different number of updates per step $I$ on ant target position task. The maximum distance was set to 5 rather than 6 for this experiment, so the numbers should be lower than the ones reported in the paper.

### A.1 GOAL STATE AND $\tau$ SAMPLING STRATEGY

While Q-learning is valid for any value of $s_g$ and $\tau$ for each transition tuple $(s_t, a_t, s_{t+1})$, the way in which these values are sampled during training can affect learning efficiency. Some potential strategies for sampling $s_g$ are: (1) uniformly sample future states along the actual trajectory in the buffer (i.e., for $s_t$, choose $s_g = s_{t+k}$ for a random $k > 0$) as in (Andrychowicz et al., 2017); (2) uniformly sample goal states from the replay buffer; (3) uniformly sample goals from a uniform range of valid states. We found that the first strategy performed slightly better than the others, though not by much. In our experiments, we use the first strategy. The horizon $\tau$ is sampled uniformly at random between 0 and the maximum horizon $\tau_{\max}$.

### A.2 MODEL-FREE SETUP

In all our experiments, we used DDPG (Lillicrap et al., 2015) as the base off-policy model-free RL algorithm for learning the TDMs $Q(s, a, g, s_\tau)$. Experience replay (Mnih et al., 2015) has size of 1 million transitions, and the soft target networks (Lillicrap et al., 2015) are used with a polyak averaging coefficient of 0.999 for DDPG and TDM and 0.95 for HER and DDPG-Sparse. For HER and DDPG-Sparse, we also added a penalty on the tanh pre-activation, as in Andrychowicz et al. (2017). Learning rates of the critic and the actor are chosen from {1e-4, 1e-3} and {1e-4, 1e-3} respectively. Adam (Kingma & Ba, 2014) is used as the base optimizer with default parameters except the learning rate. The batch size was 128. The policies and networks are parmaeterized with neural networks with ReLU hidden activation and two hidden layers of size 300 and 300. The policies have a tanh output activation, while the critic has no output activation (except for TDM, see A.5). For the TDM, the goal was concatenated to the observation. The planning horizon $\tau$ is also concatenated as an observation and represented as a single integer. While we tried various representations for $\tau$ such as one-hot encodings and binary-string encodings, we found that simply providing the integer was sufficient.

While any distance metric for the TDM reward function can be used, we chose L1 norm $-\|s_{t+1} - s_g\|_1$ to ensure that the scalar and vectorized TDMs are consistent.

### A.3 MODEL-BASED SETUP

For the model-based comparison, we trained a neural network dynamics model with ReLU activation, no output activation, and two hidden units of size 300 and 300. The model was trained to predict the difference in state, rather than the full state. The dynamics model is trained to minimize the mean squared error between the predicted difference and the actual difference. After each state is observed, we sample a minibatch of size 128 from the replay buffer (size 1 million) and perform one step of gradient descent on this mean squared error loss. Twenty rollouts were performed to compute the (per-dimension) mean and standard deviation of the states, actions, and state differences.

We used these statistics to normalize the states and actions before giving them to the model, and to normalize the state differences before computing the loss. For MPC, we simulated 512 random action sequences of length 15 through the learned dynamics model and chose the first action of the sequence with the highest reward.

### A.4    TUNED HYPERPARAMETERS

For TDMs, we found the most important hyperparameters to be the reward scale, $\tau_{\max}$, and the number of updates per observations, $I$. As shown in Figure 4, TDMs can greatly benefit from larger values of $I$, though eventually there are diminishing returns and potentially negative impact, mostly likely due to over-fitting. We found that the baselines did not benefit, except for HER which did benefit from larger $I$ values. For all the model-free algorithms (DDPG, DDPG-Sparse, HER, and TDMs), we performed a grid search over the reward scale in the range $\{0.01, 1, 100, 10000\}$ and the number of updates per observations in the range $\{1, 5, 10\}$. For HER, we also tuned the weight given to the policy pre-tanh-activation $\{0, 0.01, 1\}$, which is described in Andrychowicz et al. (2017). For TDMs, we also tuned the best $\tau_{\max}$ in the range $\{15, 25, \text{Horizon} - 1\}$. For the half cheetah task, we performed extra searches over $\tau_{\max}$ and found $\tau_{\max} = 9$ to be effective.

### A.5    TDM NETWORK ARCHITECTURE AND VECTOR-BASED SUPERVISION

For TDMs, since we know that the true Q-function must learn to predict (negative) distances, we incorporate this prior knowledge into the Q-function by parameterizing it as $Q(s, a, s_g, \tau) = -\|f(s, a, s_g, \tau) - s_g\|_1$. Here, $f$ is a vector outputted by a feed-forward neural network and has the same dimension as the goal. This parameterization ensures that the Q-function outputs non-positive values, while encouraging the Q-function to learn what we call a goal-conditioned model: $f$ is encouraged to predict what state will be reached after $\tau$, when the policy is trying to reach goal $s_g$ in $\tau$ time steps.

For the $\ell_1$ norm, the scalar supervision regresses

$$Q(s_t, a_t, s_g, \tau) = -\sum_j |f_j(s_t, a_t, s_g, \tau) - s_{g,j}|$$

onto

$$r(s_t, a_t, s_{t+1}, s_g) + \mathbb{1}[\tau = 0] + Q(s_{t+1}, a^*, s_g, \tau - 1)\mathbb{1}[\tau \neq 0]$$
$$= -\sum_j \{|s_{t+1,j} - s_{g,j}|\mathbb{1}[\tau = 0] + |f_j(s_t, a^*, s_g, \tau - 1) - s_{g,j}|\mathbb{1}[\tau \neq 0]\}$$

where $a^* = \text{argmax}_a Q(s_{t+1}, a, s_g, \tau - 1)$. The vectorized supervision instead supervises each components of $f$, so that

$$|f_j(s_t, a_t, s_g, \tau) - s_{g,j}|$$

regresses onto

$$|s_{t+1,j} - s_{g,j}|\mathbb{1}[\tau = 0] + |f_j(s_t, a^*, s_g, \tau - 1) - s_{g,j}|\mathbb{1}[\tau \neq 0]$$

for each dimension $j$ of the state.

### A.6    TASK AND REWARD DESCRIPTIONS

Benchmark tasks are designed on MuJoCo physics simulator (Todorov et al., 2012) and OpenAI Gym environments (Brockman et al., 2016). For the simulated reaching and pushing tasks, we use (8) and for the other tasks we use (9) for policy extraction. The horizon (length of episode) for the pusher and ant tasks are 50. The reaching tasks has a horizon of 100. The half-cheetah task has a horizon of 99.

*7-DoF reacher:*: The state consists of 7 joint angles, 7 joint angular velocities, and 3 XYZ observation of the tip of the arm, making it 17 dimensional. The action controls torques for each joint, totally 7 dimensional. The reward function during optimization control and for the model-free baseline is the negative Euclidean distance between the XYZ of the tip and the target XYZ. The targets are sampled randomly from all reachable locations of the arm at the beginning of each episode.

The robot model is taken from the striker and pusher environments in OpenAI Gym MuJoCo domains (Brockman et al., 2016) and has the same joint limits and physical parameters.

Many tasks can be solved by expressing a desired goal state or desired goal state components. For example, the 7-Dof reacher solves the task when the end effector XYZ component of its state is equal to the goal location, $(x^*, y^*, z^*)$. One advantage of using a goal-conditioned model $f$ as in Equation (8) is that this desire can be accounted for directly: if we already know the desired values of some components in $s_{t+T}$, then wen can simply fix those components of $s_{t+T}$ and optimize over the other dimensions. For example for the 7-Dof reacher, the optimization problem in Equation (8) needed to choose an action becomes

$$a_t = \underset{a_t, s_{t+T}[0:14]}{\operatorname{argmax}} \ r_c(f(s_t, a_t, s_{t+T}[0:14]||[x^*, y^*, z^*]))$$

where $||$ denotes concatenation; $s_{t+T}[0:14]$ denotes that we only optimize over the first 14 dimensions (the joint angles and velocities), and we omit $a_{t+T}$ since the reward is only a function of the state. Intuitively, this optimization chooses whatever goal joint angles and joint velocities make it easiest to reach $(x^*, y^*, z^*)$. It then chooses the corresponding action to get to that goal state in $T$ time steps. We implement the optimization over $s[0:14]$ with stochastic optimization: sample 10,000 different vectors and choose the best value. Lastly, instead of optimizing over the actions, we use the policy trained in DDPG to choose the action, since the policy is already trained to choose an action with maximum Q-value for a given state, goal state, and planning horizon. We found this optimization scheme to be reliable, but any optimizer can be used to solve Equation (8),(7), or (6).

*Pusher*: The state consists of 3 joint angles, 3 joint angular velocities, the XY location of the hand, and the XY location of the puck. The action controls torques for each of the 3 joints. The reward function is the negative Euclidean distance between the puck and the puck. Once the hand is near (with 0.1) of the puck, the reward is increased by 2 minus the Euclidean distance between the puck and the goal location. This reward function encourages the arm to reach the puck. Once the arm reaches the puck, bonus reward begins to have affect, and the arm is encouraged to bring the puck to the target.

As in the 7-DoF reacher, we set components of the goal state for the optimal control formulation. Specifically, we set the goal hand position to be the puck location. To copy the two-stage reward shaping used by our baselines, the goal XY location for the puck is initially its current location until the hand reaches the puck, at which point the goal position for the puck is the target location. There are no other state dimensions to optimize over, so the optimal control problem is trivial.

*Half-Cheetah*: The environment is the same as in Brockman et al. (2016). The only difference is that the reward is the $\ell$-1 norm between the velocity and desired velocity $v^*$. Our optimal control formulation is again trivial since we set the goal velocity to be $v^*$. The goal velocity for rollout was sampled uniformly in the range $[-6, 6]$. We found that the resulting TDM policy tends to "jump" at the last time step, which is the type of behavior we would expect to come out of this finite-horizon formulation but not of the infinite-time horizon of standard model-free deep RL techniques.

*Ant*: The environment is the same as in Brockman et al. (2016), except that we lowered the gear ratio to 30 for all joints. We found that this prevents the ant from flipping over frequently during the initially phase of training, allowing us to run all the experiments faster. The reward is the $\ell$-1 norm between the actual and desired xy-position and xy-velocity (for the position and velocity task) of the torso center of mass. For the target-position task, the target position was any position within a 6-by-6 square. For the target-position-and-velocity task, the target position was any position within a 1-by-1 square and any velocity within a 0.05-by-0.05 velocity-box. When computing the distance for the position-and-velocity task, the velocity distance was weighted by 0.9 and the position distance was weighted by 0.1.

*Sawyer Robot*: The state and action spaces are the same as in the 7-DoF simulated robot except that we also included the measured torques as part of the state space since these can different from the applied torques. The reward function used is also the $\ell_1$ norm to the desired XYZ position.

