# OpenReview forum: "Temporal Difference Models: Model-Free Deep RL for Model-Based Control"
_ICLR.cc/2018/Conference — Accept (Poster)_

### Official Review · AnonReviewer1 · 2017-11-26
**Nice ideas and execution; needs some more discussion of existing work**

**Rating:** 7
**Confidence:** 4

**Review:**

The paper universal value function type ideas to learn models of how long the current policy will take to reach various states (or state features), and then incorporates these into model-predictive control. This looks like a reasonable way to approach the problem of model-based RL in a way that avoids the covariate shift produced by rolling learned transition models forward in time. Empirical results show their method outperforming Hindsight Experience Replay (which looks quite bad in their experiments), DDPG, and more traditional model-based learning. It also outperforms DDPG quite a bit in terms of sample efficiency on a real robotic arm. They also show the impact of planning horizon on performance, demonstrating a nice trade-off.

There are however a couple of relevant existing papers that the authors miss referencing / discussing:
- "Reinforcement Learning with Unsupervised Auxiliary Tasks" (Jarderberg et al, ICLR 2017) - uses predictions about auxiliary tasks, such as effecting maximum pixel change, to obtain much better sample efficiency.
- "The Predictron: End-To-End Learning and Planning" (Silver et al, ICML 2017), which also provides a way of interpolating between model-based and model-free RL.

I don't believe that these pieces of work subsume the current paper, however the authors do need to discuss the relationship their method has with them and what it brings.

** UPDATE Jan 9: Updated my rating in light of authors' response and updated version. I recommend that the authors find a way to keep the info in Section 4.3 (Dynamic Goal and Horizon Resampling) in the paper though, unless I missed where it was moved to. **

---

> ### Author Response · Authors · 2018-01-05
> **Response to AnonReviewer1**
>
> Thank you for your feedback! We have edited the paper to address all of the issues that you’ve raised (see below). We would appreciate any further feedback that you might have to provide.
>
> We have added a discussion of the two papers suggested (Jaderberg et al ICML 2017, Silver et al, ICML 2017) to the paper in the related work section (Section 5, paragraph 6). Our method shares the same motivation as those papers: to increase the amount of supervision in model-free RL to achieve sample-efficient learning. We also include recent work on distributional RL (Bellemare et. al., 2017) as another example of this general idea.
>
> We were able to obtain the code for hindsight experience replay (HER) from the original authors. Using this code as reference, we improved our implementation by incorporating their hyperparameter settings and implementation details, including ones that we had difficulty deducing from the original HER paper. The performance of HER on our tasks did improve, and we have updated Figure 2 with the new results. At this point, with the help of the original authors, we are confident that our implementation of HER is accurate. An observation we would like to make is that the purpose of HER is not necessarily to improve the sample efficiency of tasks where dense rewards are sufficient for DDPG to learn. Rather, a big selling point of HER is that it can improve the asymptotic performance of DDPG in tasks with sparse rewards. To test this hypothesis, we ran an additional baseline of DDPG with sparse rewards (-1 if the goal state is not reach, 0 if it is). HER definitively outperforms this baseline, so our results confirm that HER helps with sparse-reward tasks. We wanted to improve the sample efficiency of DDPG, and not necessarily DDPG’s feasibility for sparse-reward tasks, and so we focused on tasks that DDPG could already solve, albeit after many samples. It may be that the benefits of HER shine through on tasks where dense rewards will not lead to good policies.

---

### Official Review · AnonReviewer3 · 2017-11-28
**Some interesting ideas, but not clear just how strong is the model-based/model-free connection here**

**Rating:** 4
**Confidence:** 4

**Review:**


This paper proposes a "temporal difference model learning", a method that aims to combine the benefits of model-based and model-free RL.  The proposed method essentially learns a time-varying goal-conditional value function for a specific reward formulation, which acts as a surrogate for a model in an MPC-like setting.  The authors show that the method outperforms some alternatives on three continuous control domains and real robot system.

I believe this paper to be borderline, but ultimately below the threshold for acceptance.  On the positive side, there are certainly some interesting ideas here: the notion of goal-conditioned value functions as proxies for a model, and as a means of merging model-free and model-based approaches is very really interesting, and hints at a deeper structure to goal-conditioned value functions in general.  Ultimately, though, I feel that there are two main issues that make this research feel as though it is still ultimately in the earlier stages: 1) the very large focus on the perspective that this approach is unifying model-based and model-free RL, when it fact this connection seems a bit tenuous; and 2) the rather lackluster experimental results, which show only marginal improvement over purely model-based methods (at the cost of much additional complexity), and which make me wonder if there's an issue with their implementation of prior work (namely the Highsight Experience Replay algorithm).

To address the first point, although the paper stresses it to a very high degree, I can't help but feel that the connection that the claimed advance of "unifying model-based and model-free RL" is overstated.  As far as I can tell, the connection is as follows: the learned quantity here is a time-varying goal-conditioned value function, and under some specific definition of reward, we can interpret the constraint that this value function equal zero as a proxy for the dynamics constraint in MPC.  But the exact correspondence between this and the MPC formulation only occurs for a horizon of size zero: longer horizons require a multi-step MPC for the definition of the model-free and model-based correspondence.  The fact that the action selection of a model-based method and this approach have some function which looks similar (but only under certain conditions), just seems like a fairly odd connection to highlight so heavily.

Rather, it seems to me that what's happening here is really quite simple: the authors are extending goal-conditioned value functions to the case of non-stationary finite horizon value functions (the claimed "key insight" in eq (5) is a completely standard finite-horizon MDP formulation).  This seems to describe perfectly well what is happening here, and it does also seem intuitive that this provides an advantage over stationary goal-conditioned value functions: just as goal conditioned value functions offer the advantage of considering "every state as a goal", this method can consider "every state as a goal for every time horizon".  This seems interesting enough on its own, and I admit I don't see the need for the method to be yet another claimed unification of model-free and model-based RL.

I would also suggest that the authors look into the literature on how TD methods implicitly learn models (see e.g. Boyan 1997 "Least-squares temporal difference learning", and Parr et al., 2007 "An analysis of linear models...").  In these works it has been shown that least squares TD methods (at least in the linear feature setting), implicitly learn a dynamics model in feature space, but only the "projection" of the reward function is actually needed to learn the TD weights.  In building the proposed value functions, it seems like the authors are effectively solving for multiple rewards simultaneously, which would effectively preserve the learned dynamics model.  I feel like this may be an interesting line of analysis for the paper if the authors _do_ want to stick with the notion of the method as unifying model-free and model-based RL.

All these points may ultimately just be a matter of interpretation, though, if not for the second issue with the paper, which is that the results seem quite lackluster, and the claimed performance of HER seems rather suspicious.  But instead, the authors evaluate the algorithm on just three continuous control tasks (and a real robot, which is more impressive, but the task here is still so extremely simple for a real robot system that it really just qualifies as a real-world demonstration rather than an actual application).  And in these three settings, a model-based approach seems to work just as well on two of the tasks, and may soon perform just as well after a few more episodes on the last task (it doesn't appear to have converged yet).  And despite the HER paper showing improvement over traditional policy approaches, in these experiments plain DDPG consistently performs as well or better than HER.

---

> ### Author Response · Authors · 2018-01-05
> **Response to AnonaReviewer3**
>
> Thank you for your feedback!
>
> To address the reviewer’s concerns about the experiments, we ran our algorithm on more difficult tasks and have updated the experimental results. We would like to emphasize is that a primary goal of our method is to achieve both sample-efficiency and good final performance. While the asymptotic performance of TDMs may not always be far better than that the model-free methods, TDMs are substantially more sample-efficient, as shown in the updated Figure 2. We consider this important, as sample efficiency is important in many real-world tasks, such as robotics, where collecting data is expensive. For the model-based baseline, a concern brought up was that, “model-based approach…may soon perform just as well after a few more episodes on the last task (it doesn't appear to have converged yet).” We ran the half-cheetah experiments for more iterations and see that the trend is the same: TDM converges to a better solution than the model-based baseline, by a significant margin. We also evaluated our method on two substantially more complex 3D locomotion tasks using the “ant” quadrupedal robot. We tested two tasks: asking the ant to run to a target location, and asking it to achieve a target location at the same time as a target velocity (this is meant to be representative, e.g., of the ant attempting to make a jump). The latter task is particularly interesting, since the ant cannot maintain both the position and velocity goal at the same time, and must therefore achieve it at a single time step. TDMs significantly outperform the model-based baseline on these tasks as well.  This supports the central claim of the paper, in Section 1, paragraph 5, which is that TDMs achieve learning times comparable to model-based methods, but asymptotic performance that is comparable to model-free algorithms. We believe that this kind of improvement in learning performance is of substantial interest to the reinforcement learning community.
>
> We were able to obtain the code for hindsight experience replay (HER) from the original authors. Using this code as reference, we improved our implementation by incorporating their hyperparameter settings and implementation details, including ones that we had difficulty deducing from the original HER paper. The performance of HER on our tasks did improve, and we have updated Figure 2 with the new results. At this point, with the help of the original authors, we are confident that our implementation of HER is accurate. An observation we would like to make is that the purpose of HER is not necessarily to improve the sample efficiency of tasks where dense rewards are sufficient for DDPG to learn. Rather, a big selling point of HER is that it can improve the asymptotic performance of DDPG in tasks with sparse rewards. To test this hypothesis, we ran an additional baseline of DDPG with sparse rewards (-1 if the goal state is not reach, 0 if it is). HER definitively outperforms this baseline, so our results confirm that HER helps with sparse-reward tasks. We wanted to improve the sample efficiency of DDPG, and not necessarily DDPG’s feasibility for sparse-reward tasks, and so we focused on tasks that DDPG could already solve, albeit after many samples. It may be that the benefits of HER shine through on tasks where dense rewards will not lead to good policies.
>
> We appreciate your comments regarding the connection between model-based and model-free RL. In this paper, we presented two main contributions: one is a connection between model-free and model-based reinforcement learning, and another is an algorithm derived from this connection. We have edited the paper (throughout the paper, and notably in Section 3.2, Paragraph 2) to balance the presentation better between these two components, and to avoid overstating the connection. We would be happy to incorporate any other concrete suggestions you might have.
>
> Thank you for the references to the earlier work connecting TD-methods and model-based methods. We have added a discussion to this work in the related works (Section 5, paragraph 2). While these papers also show a connection between TD-methods and model-based methods, their objective is rather different from ours. Boyan shows an exact equivalence between a learned model and learned value function, but this requires a tabular value function which effective keeps track of every state-action-next-state transition. Parr shows that for linear function approximators, a value function extracted using a learned model is the same as a value function learned with TD-learning. Rather than analyzing equivalence at convergence, our primary contribution is how we can achieve sample complexity comparable to model-based RL while retaining the favorable asymptotic performance of model-free RL in complex tasks with function approximation.
>
> We believe that we have addresses all the issues raised by the reviewer. We would be happy to discuss and address any additional concerns.

---

### Official Review · AnonReviewer2 · 2017-11-30
**interesting direction**

**Rating:** 7
**Confidence:** 3

**Review:**

This is an interesting direction. There is still much to understand about the relative strengths and limitations of model based and model free techniques, and how best to combine them, and this paper discusses a new way to address this problem. The empirical results are promising and the ablation studies are good, but it also makes me wonder a bit about where the benefit is coming from.

Could you please put a bit of discussion in about the computational and memory cost. TDM is now parameterized with (state, action (goal) state, and the horizon tau). Essentially it is now computing the distance to each possible goal state after starting in state (s,a) and taking a fixed number of steps.
It seems like this is less compact than learning a 1-step dynamics model directly.
The results are better than models in some places. It seems likely this is because the model-based approach referenced doesn’t do multi-step model fitting, but essentially TDM is, by being asked to predict and optimize for C steps away. If models were trained similarly (using multi-step loss) would models do as well as TDM?
How might this be extended to the stochastic setting?

---

> ### Author Response · Authors · 2018-01-05
> **Response to AnonReviewer2**
>
> Thank you for your feedback!
>
> One concern brought up is the computation and memory cost of using TDMs. To address this concern, we have added discussion of this point in Section 4.3, paragraph 2, as well as Figure 4 in the appendix. In short, the learning for TDM and DDPG both have the number of updates per environment step as a hyperparameter, which largely determines the computation cost. The empirical result we got is that as we increased the number of updates, the performance of TDM increased while the performance of DDPG stayed the same or degraded. TDMs can benefit from more computation (number of updates per environment step) than DDPG since they can learn a lot more by relabeling goal states and horizon tau; we see this more as a benefit as it means TDMs can extract more information from the same amount of data. We also would like to point out that one advantage of doing more computation at training time is that test time is relatively fast: to do multi-step planning, we simply set tau=5 in our TDM, whereas a typical multi-step model-based planning approach would need to unroll a model over five time steps and optimize over all intermediate actions. Furthermore, we hope that this, in addition to the ablative studies in section 6.2, addresses the concern that, “… it also makes me wonder a bit about where the benefit is coming from.”
>
> For stochastic environments, the TDM would learn the expected distance to the goal, rather than the exact distance. We have added this discussion to the second paragraph of the conclusion.
>
> We agree that a more in-depth discussion of the connection to multi-step models would be appropriate. We’ve added discussion of two related works in Section 5, paragraph 4. One critical distinction between these methods and TDMs is that TDMs can be viewed as goal conditioned models: the prediction is made T steps into the future, conditioned on a policy that is trying to reach a particular state. Most model learning methods do not condition on a policy, requiring them to take in an entire sequence of future actions, which greatly increases the input space.

---

### Public Comment · (anonymous) · 2018-02-25
**some questions**

You chose DDPG as the model-free baseline while DDPG is very sensitive to initial parameters, and would you mind comment why the model-based baseline is not a multi-step prediction [1]? Would you mind open source your code?

[1] Neural Network Dynamics for Model-Based Deep Reinforcement Learning with Model-Free Fine-Tuning. Anusha Nagabandi, Gregory Kahn, Ronald S. Fearing, Sergey Levine

---

> ### Author Response · Authors · 2018-02-27
> **DDPG, Model-based baseline, and open-source code**
>
> Our experience with DDPG has also been that it is sensitive to hyperparameters, and so we performed a hyperparameter sweep for all the tasks (See Appendix, section A.4).
>
> Code is available at https://github.com/vitchyr/rlkit, though it currently only has the implementation that uses Equation (9) to extract a policy.
>
> TDMs are only trained with tuples of state-action-next-state transitions, and our model-based baseline also does not train on sequences of actions and states. We agree that it would be interesting to see how multi-step model-based methods would perform, though it may become difficult to establish a fair way to compare the methods since TDMs (as presented in this paper) are only trained with sequences of length one.

---

### Decision · Program_Chairs · 2018-01-29
**ICLR 2018 Conference Acceptance Decision**

**Decision:**

Accept (Poster)

**Comment:**

There is a concern from one of the reviewers that the paper needs deeper analysis. On the other hand, applying finite horizon techniques to deep RL is relatively unexplored, and the paper does provide some interesting results in that direction.